# EEG Classification of Normal and Alcoholic by Deep Learning

**DOI:** 10.3390/brainsci12060778

**Published:** 2022-06-14

**Authors:** Houchi Li, Lei Wu

**Affiliations:** 1School of Computer Science and Engineering, Hunan University of Science and Technology, Xiangtan 411100, China; 18175681159@163.com; 2Hunan Engineering Research Center for Intelligent Decision Making and Big Data on Industrial Development, Hunan University of Science and Technology, Xiangtan 411100, China

**Keywords:** alcoholism, machine learning, EEG signals, discrete wavelet transform, bidirectional long short-term memory, convolutional neural network

## Abstract

Alcohol dependence is a common mental disease worldwide. Excessive alcohol consumption may lead to alcoholism and many complications. In severe cases, it will lead to inhibition and paralysis of the centers of the respiratory and circulatory systems and even death. In addition, there is a lack of effective standard test procedures to detect alcoholism. EEG signals are data obtained by measuring brain changes in the cerebral cortex and can be used for the diagnosis of alcoholism. Existing diagnostic methods mainly employ machine learning techniques, which rely on human intervention to learn. In contrast, deep learning, as an end-to-end learning method, can automatically extract EEG signal features, which is more convenient. Nonetheless, there are few studies on the classification of alcohol’s EEG signals using deep learning models. Therefore, in this paper, a new deep learning method is proposed to automatically extract and classify EEG’s features. The method first adopts a multilayer discrete wavelet transform to denoise the input data. Then, the denoised data are used as input, and a convolutional neural network and bidirectional long short-term memory network are used for feature extraction. Finally, alcohol EEG signal classification is performed. The experimental results show that the method proposed in this study can be utilized to effectively diagnose patients with alcoholism, achieving a diagnostic accuracy of 99.32%, which is better than most current algorithms.

## 1. Introduction

Excessive alcohol consumption is the most common mental illness in the world, and deaths from alcohol intoxication due to excessive alcohol consumption frequently occur. According to the WHO global status report on alcohol and health 2018, the harmful use of alcohol resulted in 3.3 million deaths worldwide, accounting for approximately 5.9% of total global deaths [1]. Approximately 2 billion people around the world consume alcoholic beverages, and most of them suffer from alcoholism. It has been estimated that alcohol abuse represents the world’s third-largest risk factor for disease and disability [2]. Long-term alcohol use not only causes damage to many organs, such as the liver, gallbladder and heart muscle, but also causes irreversible damage to the nervous system, resulting in mental health problems and memory loss [3,4].

Excessive alcohol consumption in a short period of time can lead to symptoms such as dyskinesia, stumbling, incoherence, and increased heart rate. If alcoholism is not detected and treated at an early stage, it may lead to other complications, such as accidental injury, acid-base imbalance, water and electrolyte disorders, pneumonia and even acute renal failure. These complications will cause serious damage to the patient’s health and endanger the patient’s life. In addition, alcohol not only affects alcoholics but also causes serious harm to people and society because of the behaviors of the person with alcoholism. The diagnosis of alcoholism is currently complicated by the lack of objective criteria. The assessment of alcohol abuse and alcoholism is usually based on, but not limited to, the following activities: (i) Question the patient on their alcohol drinking history. (ii) Perform a physical exam and complete a psychological evaluation. (iii) Perform lab tests, such as the detection of serum ethanol concentration and arterial blood gas analysis. However, such procedures are laborious, time-consuming, and subjective and thus may lead to diagnostic errors. Considering that most of the harmful consequences of alcoholism are based on altered brain function, an electroencephalogram (EEG), which reflects the status of the brain, can be used as a noninvasive and objective technique to assist in the diagnosis of alcoholism [5,6]. Taran S et al. proposed an automatic recognition and analysis method of alcohol EEG signals based on the characteristics of brain rhythm [7]. Mumtaz W et.al. summarized EEG signal abnormalities associated with the condition of patients with alcohol use disorders [8]. Bavkar S proposed a rapid screening method for alcoholism based on EEG signals [9].

In recent years, extensive research has been performed on the use of EEG signals to diagnose alcoholism. The traditional method for diagnosing alcohol EEG signals is to determine whether alcohol intoxication is caused by visual analysis of EEG signal data by experienced medical experts based on their own clinical experience, which requires much time and effort [10]. With the development of computer technology, computer-aided diagnosis technology has been applied to the diagnosis of EEG signals. Compared with traditional diagnosis methods, computer-aided EEG analysis technology provides a faster and more effective method. This technique reduces the errors that may be caused by manual diagnosis and is more objective [8,9,10,11,12,13,14,15,16,17,18,19,20,21,22]. Interference during the acquisition of EEG signals is usually caused by eye blinks, head movements, and EMG activities of muscles near the recording point, resulting in artifact contamination. Therefore, in computer-aided diagnosis, it is usually necessary to extract the features of EEG signals to improve the signal-to-noise ratio [12,13]. The commonly used feature extraction methods can be summarized as time domain analysis [14,15], frequency domain analysis [16,17] and time-frequency domain analysis [18,19]. Since EEG signals are usually transient and unstable, compared with time-domain analysis and frequency-domain analysis, time-frequency domain analysis has the ability to represent the local characteristics of signals in both the time domain and frequency domain and has more advantages in processing nonstationary signals. Common time-frequency analysis methods include empirical mode decomposition (EMD) [20,21], wavelet transform [22], short-time Fourier transform (STFT) [23] and so on. Machine learning technology has been widely used as a standard procedure to evaluate the meaning of EEG signals. Machine learning combines various feature extraction methods with classifiers for EEG signal processing and alcoholism detection [24,25,26,27,28,29,30,31,32,33]. However, the traditional feature extraction of machine learning mainly depends on manual extraction. For simple and obvious tasks, manual feature extraction is simple and effective, but when the features of EEG signals are not obvious, it is often difficult for hidden features to be correctly extracted.

Deep learning, which automatically learns from complex datasets, can be utilized to discover hidden features that are difficult to find manually. Currently, there are some related works on the diagnosis of alcohol EEG signals based on deep learning. Commonly used methods are ANN, CNN, and LSTM networks [34,35,36,37]. However, there is still a lack of effective deep learning systems to diagnose alcoholism through EEG signals. Therefore, this research aims to fill this gap, and a DWT-CNN-Bi-LSTM deep learning framework is proposed. In this research, we adopt a discrete wavelet transform to extract the time-frequency domain features of the signal, then use the convolutional neural network and bidirectional long short-term memory network to further extract the spatiotemporal features in the signal, and finally accurately classify these features as a patient with alcoholism or patient control. Compared with machine learning methods, the architecture we propose can automatically extract hidden features in EEG signals. In addition, the EEG signals have temporal and spatial information. Most of the current deep learning methods only extract temporal or spatial information, while our deep learning architecture captures the temporal and spatial features in EEG signals. At the same time, the Bi-LSTM network can capture the information of the network before and after and improve the accuracy of the experiment. The experimental results show that the classification accuracy of the model reaches 99.32%, which can be used in the diagnostic classification system of alcoholism.

## 2. Materials and Methods

### 2.1. Dataset and Configurations

The dataset used in this research was collected by the Neurodynamics Laboratory at the State University of New York Health Center in Brooklyn and was taken from the machine learning library of the University of California, Irvine (UCI) [38]. In this research, we adopted the full dataset. The data in this dataset contain measurements from 64 electrodes placed on the scalp of the subject. In this research, each electrode represents a channel, and the input terminal takes a total of 64 channels of EEG signal data. These electrodes were sampled at 256 Hz for 1 s. A total of 120 experiments were collected from 122 subjects, and different stimuli were utilized. The electrode positions were located at standard sites (Standard Electrode Position Nomenclature, American Electroencephalographic Association 1990). As shown in Figure 1, sample a and sample b represent alcohol EEG signals and control EEG signals, respectively. To evaluate the performance of the proposed DWT-CNN-Bi-LSTM system, our implementation was derived in Python using Keras with TensorFlow backend, and one hour of training was completed using an NVIDIA 1080 GPU.

### 2.2. StandardScaler

In the electrical signal analysis experiment, sometimes there are sometimes few samples available for a single subject. To enhance the credibility of the classification, it is necessary to uniformly classify multiple subjects’ eigenvalues uniformly. The characteristic values of EEG signals of different subjects are usually different, and sometimes there are even great differences in the order of magnitude, which will lead to a decrease in classification accuracy [39]. Therefore, this research uses standard deviation standardization to deal with the initial data. The standardized data are subtracted from the mean and then divided by the standard deviation. The processed data are in accordance with the standard normal distribution, i.e., the mean is 0, and the standard deviation is 1. The standardization equation is as follows: (1)X=(X−μ)/σ
where μ is the mean value of all sample data, and σ is the standard deviation of all sample data.

### 2.3. Discrete Wavelet Transform

At the same time, due to the nonstationary and transient characteristics of EEG time-series signals as well as artifacts caused by electromagnetic interference and muscle movement, it is difficult for signal analysis based on fast Fourier transform to capture the instantaneous frequency of events. As a time-frequency analysis method of the signal, wavelet transform can be used to represent the local characteristics of the signal in the time-frequency domain. The continuous wavelet transform formula is defined as follows: (2)$$W_{f}\left(a,b\right)=\frac{1}{\sqrt{\left|a\right|}}\int_{-\infty }^{+\infty }f\left(t\right)\psi \left(\frac{t-b}{a}\right)dt$$
where *a* and *b* are scale parameters and time parameters or translation parameters, respectively. *W_f_*(*a*,*b*) represents the transformation value of the signal *f*(*t*) after wavelet transform when the frequency is *a* and the time is *b*. However, the amount of calculation required for continuous wavelet transform is very large, and the calculated results are redundant to a certain extent. To simplify the calculation without losing the result information, the discrete wavelet transform of a continuous signal is implemented. The DWT is defined as: (3)$$D\left(j,k\right)=2^{-}\left(j/2\right)\int_{-\infty }^{+\infty }x\left(t\right)\psi \left(\frac{t-2^{j}k}{2^{j}}\right)dt$$

Scaling and shifting parameters are converted to powers of two, called dyadic scales and positions, respectively. Among them, a and b are replaced by $2^{j}$ and $k2^{j}$, respectively, Ref [40]. In this study, a discrete wavelet transform is used to process the standardized data, and the approximate signal is continuously decomposed to improve the signal-to-noise ratio of the brain signal to obtain higher quality data.

### 2.4. Convolutional Neural Network

In recent years, deep learning has been greatly developed and widely used in various fields. Convolutional neural networks, which are among the most well-known deep learning models, extract rich features by using various filters in the convolution layer, pooling layer, normalization layer and fully connected layer to improve the performance of various tasks [41]. By using multiple filters to perform a one-dimensional convolution operation, a one-dimensional convolutional neural network can extract effective and representative features from time-series data. Moreover, because the convolution filter and feature mapping adopted in this research are one-dimensional, a one-dimensional measurement data sequence can be directly used as the input of a one-dimensional convolutional neural network without conversion to a two-dimensional signal, maintaining the correlation of the original signal.

### 2.5. Bidirectional Long Short-Term Memory

Although convolutional neural networks show great advantages in feature extraction, they cannot retain the memory of previous time-series patterns. Compared with convolutional neural networks, LSTM networks are more successful in processing time data. The principle of the LSTM hidden layer structure is shown in Figure 2. $f_{t}$, $i_{t}$ and $o_{t}$ represent the values of the forget gate, input gate and output gate at time *t*, respectively, and at represents the preliminary feature extraction of $h^{t-1}$ and $x_{t}$ at time *t* [42].

The mathematical expression of the LSTM unit is defined as follows: (4)$$f\left(t\right)=\sigma \left(W_{f}h^{t-1}+U_{f}x_{t}+b_{f}\right)$$
(5)$$i\left(t\right)=\sigma \left(W_{i}h^{t-1}+U_{i}x_{t}+b_{i}\right)$$
(6)$$a\left(t\right)=tanh\left(W_{a}h^{t-1}+U_{a}x_{t}+b_{a}\right)$$
(7)$$o\left(t\right)=\sigma \left(W_{o}h^{t-1}+U_{o}x_{t}+b_{o}\right)$$
where $x_{t}$ represents the input of moment *t*, $h^{t-1}$ represents the hidden layer state value of moment *t*; $W_{f}$, $W_{i}$, $W_{o}$ and $W_{a}$ represent the weight coefficients of $h^{t-1}$ in the process of forget gate, input gate, output gate and feature extraction, respectively; $U_{f}$, $U_{i}$, $U_{o}$ and $U_{a}$ represent the weight coefficient of $x_{t}$ in the process of forget gate, input gate, output gate and feature extraction, respectively. $b_{f}$, $b_{i}$, $b_{o}$ and $b_{a}$ represent the offset values of the forget gate, the input gate, the output gate and the feature extraction process, respectively; tanh represents the tangent hyperbolic function, and σ represents the sigmoid activation function. The results of the calculation of the forget gate and the input gate act on c(t−1) to form the cell state c(t) at time *t*, which is expressed by the formula as follows: (8)c(t)=c(t−1)⊙f(t)+i(t)⊙a(t)
where ⊙ is the Hadamard product. Finally, the hidden layer state h(t) at time *t* is calculated from the output o(t) and the current cell state c(t): (9)h(t)=o(t)⊙tanh(c(t))

Generally, the LSTM network can only obtain information from the previous input, while the bidirectional long short-term memory (Bi-LSTM) model, as a cyclic neural network composed of two independent LSTM networks, can obtain information not only from the previous input but also from the future input state. The model design idea of Bi-LSTM is to make the characteristic data obtained at moment t have the information between the past and the future at the same time, and its principle structure is shown in Figure 3.

### 2.6. CNN-Bi-LSTM Model

The CNN-Bi-LSTM structure model proposed in this paper consists of one input layer, four convolution layers, two max pooling layers, four fully connected layers and one sigmoid output layer. At the same time, to prevent overfitting, a dropout layer is added between each fully connected layer to improve the generalization ability of the model. The data processed by the discrete wavelet transform are used as the input of the model, and the abstract features of the data are extracted by a convolutional neural network. Figure 4 shows the input of the bidirectional long short-term memory network (Bi-LSTM). These networks can cooperate with each other, retain the previous information, and further improve the ability to learn useful information from the EEG time-series data. After passing through the Bi-LSTM layer, the output features will be sent to three fully connected layers. Finally, the sigmoid output layer is added to the model for final recognition. The specific experimental steps are as follows. The shape of the input data (76 × 1) is obtained by discrete wavelet transform. Then, through the first convolutional layer, the abstract features of the input signal are extracted, and the number of one-dimensional convolution kernels in the Conv layer is 64. The shape of each convolution core is 3 × 1, and the stride of the convolution kernel is 1. After the convolutional layer is a rectified linear unit (ReLU) activation layer, which is added to the network to enhance the representation ability of the network and solve problems that cannot be solved by linear models. After the convolution and activation layers, the output will go through a max pooling layer to reduce dimensionality, remove redundant information, compress features, simplify network complexity, and reduce overfitting. The pooling window size is 2, and the step size is 2. Then, the higher-level features are further extracted through a three-layer convolution. The number of convolution kernels in Conv layer 1, Conv layer 2, and Conv layer 3 are 64, 128, and 128, respectively, all of which use the same 3 × 1 shape, and the activation function is the same as above. After layer 3, we add a max pooling layer. Finally, the resulting features are mapped to a fully connected layer with 256 neurons. Then, dropout is added to the fully connected layer. As shown in Figure 5, the fully connected layer plays the role of mapping the extracted distributed feature representation to the sample label space as the input of the Bi-LSTM network, and dropout can alleviate the overfitting problem to a certain extent. Both Bi-LSTM networks have 64 neurons. After the features are extracted by the Bi-LSTM network, the output signal is sent to three fully connected layers, and the hidden layers have 256, 128, and 64 neurons. Finally, the sigmoid output layer is added to the model for final recognition. To make the model output reach the optimal value, we use the Adam optimization algorithm to update and calculate the network parameters that affect the model training and output. The learning rate is 0.001, the number of epochs is 100, and the batch size is 200. An abstract illustration of the proposed approach is shown in Figure 6. First, the original signal is preprocessed, and then the discrete wavelet transform is used to decompose layer by layer to improve the signal-to-noise ratio. Next, a one-dimensional convolutional neural network and a bidirectional LSTM network are used to extract spatiotemporal features, and finally, classification is performed.

### 2.7. Model Performance Evaluation

In this study, four evaluation indicators are used to evaluate the performance of the architecture: accuracy, F1-score, precision and recall. These performance metrics are briefly described below: (10)$$\mathrm{Accuracy}=\frac{TP+TN}{TP+FP+TN+FN}$$

Accuracy represents the ratio of the number of correct decisions to all the decisions.
(11)$$\mathrm{Precision}=\frac{TP}{TP+FP}$$

Precision refers to the proportion of correctly predicted positive samples to all predicted positive samples.
(12)$$\mathrm{Recall}=\frac{TP}{TP+FN}$$

Recall, also known as sensitivity or hit rate, refers to the proportion of positive samples that are correctly predicted.
(13)$$\text{F1-score}=\frac{2(\mathrm{Precision}*\mathrm{Recall})}{\left(\mathrm{Precision}+\mathrm{Recall}\right)}$$

The F1-score is an indicator of the comprehensive consideration of accuracy and recall rate.

## 3. Experimental Results and Analysis

To improve the generalization performance of the model and avoid the overfitting problem, dropout technology is introduced into the model, and the Adam optimizer is used to alleviate the gradient oscillation problem. The changes in training accuracy and training loss with the increase in epochs are shown in Figure 7. In this figure, the left vertical axis is the training accuracy, the right vertical axis is the training loss, the red curve is the training accuracy, and the blue curve is the training loss. Moreover, the accuracy and loss curves converge faster, the frustration is smaller, and the curves are smoother, which proves that our model has high robustness in this dataset.

Table 1 shows the comparison of our model with CNN and the LSTM and Bi-LSTM networks using performance indicators. From this table, we can see that the accuracy, precision, recall and F1-score achieved by our model are 99.32%, 99.01%, 98.87%, 98.93%, respectively, which are significantly better than those of the CNN [43], LSTM [44], Bi-LSTM [45], and CNN+Bi-LSTM models. Improvements of 2.37%, 7.71%, 11.08% and 2.03%, respectively, are observed when using this model. Among the CNN, LSTM, and Bi-LSTM models, CNN achieved the best classification accuracy of 96.95% because of its powerful feature extraction ability. Compared with the LSTM network, the Bi-LSTM network has the ability to consider the information from the front and back networks, so it has higher classification accuracy. We also compare the machine learning methods. As shown in Table 2, the classification accuracy of XGBoost [46] is 79.58%, the classification accuracy of graded CatBoost [47] is 94.14%, the classification accuracy of random forest (RF) [48] is 87.98%, and the classification accuracy of the support vector machine (SVM) [24] is 95.63%. The classification accuracy of the K-nearest neighbor (KNN) algorithm [27] is 94.23%. Our method is more robust than traditional machine learning methods.

Table 3 shows a comparison of our proposed DWT-CNN-Bi-LSTM architecture with existing technologies. Most of the existing models are based on machine learning. These machine learning models use handmade features, which require domain knowledge and are difficult to apply. Farsi et al. proposed a deep learning method based on an LSTM network to automatically extract time-series information from EEG data. However, there is no suitable mechanism for dealing with spatial information. Our proposed CNN-Bi-LSTM model can automatically extract hidden spatiotemporal information from EEG signals, which has more advantages than some existing machine learning and deep learning methods.

## 4. Conclusions and Prospects

In this paper, a deep learning framework integrated with the DWT, CNN and Bi-LSTM network is proposed and used to automatically extract and classify the time-series features of alcohol EEG signals. In this method, the data are preprocessed through standard deviation standardization and discrete wavelet transform, and then the denoised data are input into the CNN-Bi-LSTM network. The average classification accuracy of this method is 99.32%, the accuracy is 99.01%, and the recall rate is 98.87%. This mechanism is superior to the existing multichannel EEG signals in predicting alcoholism and has high accuracy and reliability. The experimental results show that automatic feature extraction from EEG signals by deep learning has more advantages than manual feature extraction. In recent years, with the development of the Internet of Things and the improvement of people’s economic level, an increasing number of people have begun to pay attention to their own health problems. Medical wearable devices can upload the wearer’s body information in real-time and respond in time to ensure the health of patients. The deep learning strategy proposed by the study can provide a new and powerful diagnostic scheme for medical wearable devices. Similarly, our proposed deep learning model can also be extended to epilepsy diagnosis, emotion recognition and other applications through EEG classification. It is also an effective deep learning model for signals such as ECG and EMG.

## Figures and Tables

**Figure 1 brainsci-12-00778-f001:**
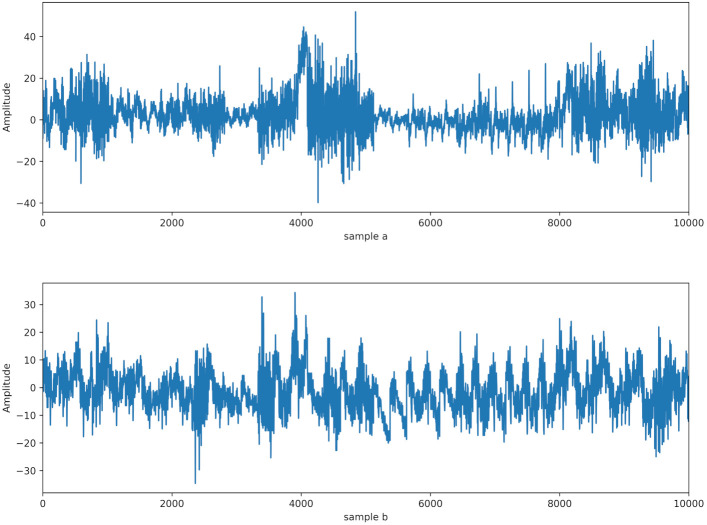
Sample (**a**) and sample (**b**) are alcoholic and normal EEG signals, respectively.

**Figure 2 brainsci-12-00778-f002:**
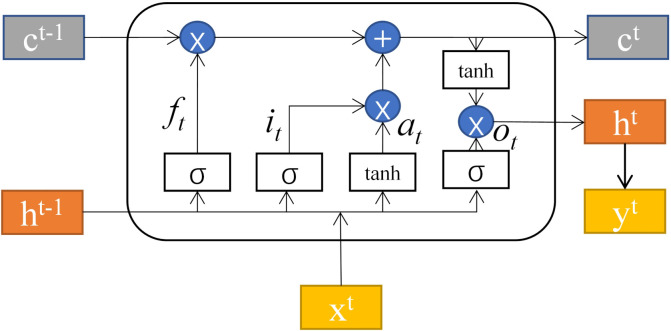
LSTM network structure.

**Figure 3 brainsci-12-00778-f003:**
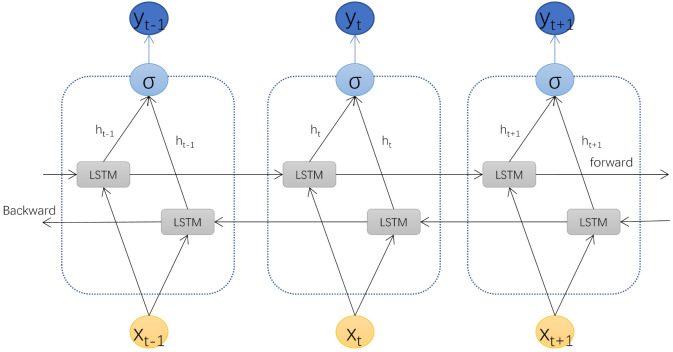
BI-LSTM network structure.

**Figure 4 brainsci-12-00778-f004:**
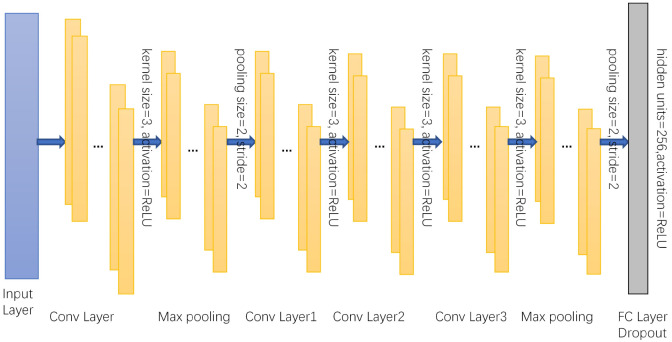
CNN network structure.

**Figure 5 brainsci-12-00778-f005:**
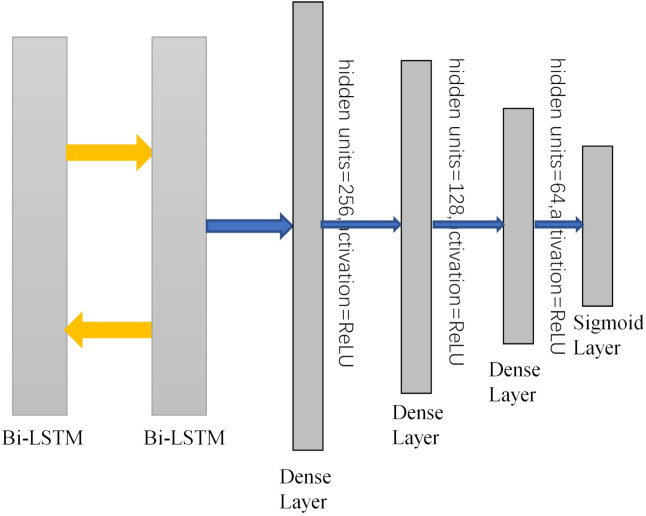
CNN network structure.

**Figure 6 brainsci-12-00778-f006:**
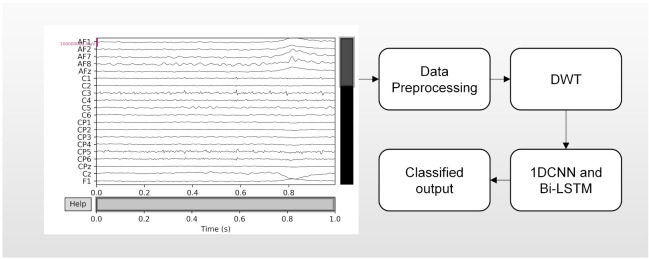
CNN network structure.

**Figure 7 brainsci-12-00778-f007:**
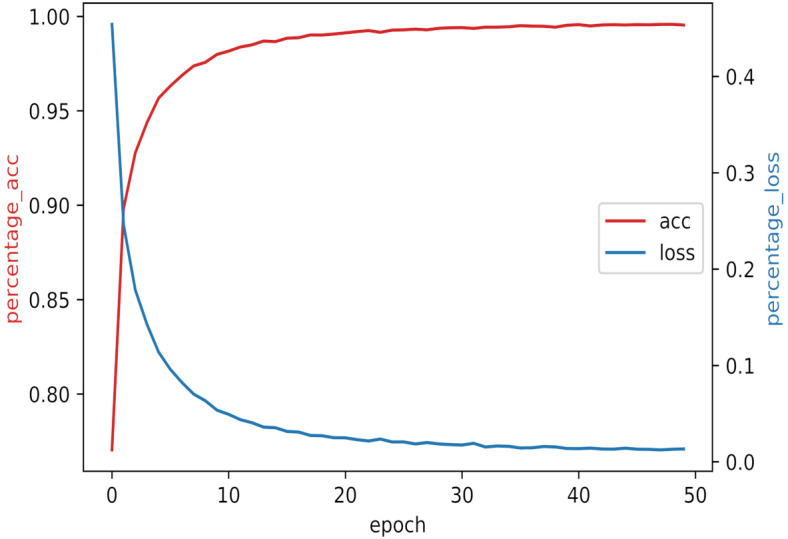
Training accuracy and training loss.

**Table 1 brainsci-12-00778-t001:** Performance of different deep learning models in binary classification tasks.

Methods	Accuracy	Precision	Recall	F1-Score
CNN	96.95%	96.05%	95.70%	96.58%
Bi-LSTM	91.61%	91.60%	90.41%	91.39%
LSTM	88.24%	88.42%	87.98%	89.21%
CNN+Bi-LSTM	97.29%	96.89%	97.13%	96.93%
Our model	99.32%	99.01%	98.87%	98.93%

**Table 2 brainsci-12-00778-t002:** Mainstream classification methods and comparison of our proposed models.

Comparison of Different Models						
Model	XGBoost	CatBoost	RF	SVM	KNN	Our Model
Accuracy	79.58%	94.14%	87.98%	95.63%	94.23%	99.32%

**Table 3 brainsci-12-00778-t003:** Comparison of DWT-CNN-Bi-LSTM architecture with existing methods.

Reference	Feature Extraction	Classifier	Accuracy
Acharya et al. [24]	Approximate Entrop LLE Sample Entropy Higher Order Spectra	SVM	91.7%
Faust et al. [27]	Wavelet Packet Decomposition	K-NN	95.8%
Patidar et al. [31]	Tunable-Q Wavelet Transform Centered Correntropy PCA	LS-SVM	97.02%
Farsi et al. [34]	Improved Binary Principal Component Analysis	ANN LSTM	93%
Sharma et al. [49]	Three-band Orthogonal Wavelet Filter Bank	LS-SVM	97.08%
Ildar et al. [35]	Wavelet transforms	CNN	86%
Bavkar et al. [50]	Linear Nonlinear Statistical Feature	K-NN	98.25%
Mukhtar H et al. [36]	CNN	CNN with 3 convolution layers, dropout	98.00%
N Kumari et al. [37]	None	CNN LSTM CNN + LSTM	92.77% 89% 91%
Our model	Discrete Wavelet Transformation	CNN Bi-LSTM	99.32%

## Data Availability

All data related to the research are presented in the article.

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
