# Peer review of "EEG Classification of Normal and Alcoholic by Deep Learning"

_brainsci, 2022, doi:10.3390/brainsci12060778_

Round 1

Reviewer 1 Report

This article addresses the classification of EEG brain signals using deep learning for the detection of alcoholism from the signals.

The problem itself is old and there are several dozens of approaches using the same dataset with comparable results. As such this is yet another study with a slightly different adoption of a mix of techniques.

The proposed method combines existing feature extraction and transformation techniques such as scaling,  discrete wave transformation, and CNN with BiLSTM.

I see some issues with their work.

1. They did not describe exactly which channels have been used in the experiment.

2. They did not detail and regularization or fine-tuning required by the network. Did they just arrange a group of layers and it worked out automatically? What would have happened if the layers/parameters were changed? Someone who wants to replicate or improve your work needs the details which are missing here.

3. Tables 1 and 2 provide a comparison of the "proposed" model with other techniques, yet it is not clear from where do these other techniques come? Are they implemented by the authors as well or are they taken from the literature? In the latter case, references must be given. 

4. Did you do any data cleaning or just took some dataset and started applying your method on that? Which dataset copy exactly was used? There are many out there with different preprocessing on the original data. 

Would it be possible to make your dataset/code available to others?

It would be better if the work could be explained as a pipeline: feature extraction, transformation, deep learning application,...

I also recommend that the author emphasize on the novelty brought in this work.

The authors are missing on some recent work in EEG classification using deep learning, e.g., in [1][2].

[1] Praveena et al. Deep Learning Techniques for EEG Signal Applications – a Review, IETE Journal of Research, 2020.

[2] Mukhtar et al. Deep Convolutional Neural Network Regularization for Alcoholism Detection Using EEG Signals, Sensors, 2021.

Author Response

Thank you for your suggestions, all your suggestions are very important, they have important guiding significance for my thesis writing and scientific research work! we amended the relevant part in manuscript. Some of your questions were answered below:

  • We supplemented the channel description in the dataset, using a total of 64 electrodes, each electrode represents a channel, and the input terminal takes a total of 64 channels of EEG signal data. e.g., page 3, line 101.
  • In Section 2.6, we supplement the network details. e.g., page 6, line 211.
  • In the experimental section, we compared other techniques used on this dataset and provided references for these techniques in Table 1 and Table 2. e.g., in [43-48].
  • The dataset is public. We use the full data set, which are described in detail in the Dataset section. The innovative points of our work are supplemented in the introduction section, e.g., page2, line 84-90, and we supplement the relevant literature on deep learning. g., in [34-37]. At the same time, we made a graphic summary, in figure 6. e.g., Page 7.

The revised manuscript and responses to each of your suggestions are in the attachments, please check them out, thanks again for your suggestions. Thank you very much for all your help and looking forward to hearing from you soon.

Best regards

Sincerely yours

Mr. Wu

Reviewer 2 Report

The authors initiated a study in a very interesting and a much helpful area in the healthcare segment. However, the manuscript can be improved in the following sections-

1. English is a major problem in this manuscript right from its beginning. MAny hanging and incomplete sentences are observed in this manuscript; which decreases the readability of the manuscript. Careful revisions are strongly recommended.

2. Literature Review:

Authors missed out on many important citations in the selected area of study such as-

a)  Rhythm-based identification of alcohol EEG signals (DOI: https://doi.org/10.1049/iet-smt.2017.0232)

b) Rapid Screening of Alcoholism: An EEG Based Optimal Channel Selection Approach (doi:10.1109/ACCESS.2019.2927267)

c) Optimal EEG channels selection for alcoholism screening using EMD domain statistical features and harmony search algorithm(https://doi.org/10.1016/j.bbe.2020.11.001)

These and such similar citations must be the part of literature review to get completeness.

3.  Table 3 shows a marginal improvement in the results i.e. 1% as compared to the well-known reported methods stated in Ref. No. 35,37 which can not prove the novelty of the reported method in this manuscript. It is suggested that authors must include a detailed qualitative comparative analysis of the proposed method with in-depth discussion.

4. Authors are suggested to include the graphical abstract of the proposed method in an appropriate section of the manuscript with its brief discussion.

Author Response

Thank you for your suggestions, all your suggestions are very important, they have important guiding significance for my thesis writing and scientific research work! we amended the relevant part in manuscript. Some of your questions were answered below:

  • We asked native English speakers to revise the article and issued a document certifying that the manuscript was edited by one or more highly qualified native English-speaking editors.
  • We cite the literature you advised in the article, and also supplement the relevant deep learning literature.[7] [9] [21] [34-37]
  • For the comparison of the methods in Table 3, our method is a deep learning method, which can automatically extract hidden features, which is more convenient than the machine learning method. At the same time, our method can extract the temporal and spatial features in the EEG signals, and save the information of the before and after network. The innovative nature of our method is supplemented in the introduction. g., page 2.
  • We made a graphic summary with its brief discussion in section II. g., page 8.

The revised manuscript and responses to each of your suggestions are in the attachments, please check them out, thanks again for your suggestions. Thank you very much for all your help and looking forward to hearing from you soon.

Best regards

Sincerely yours

Mr. Wu

Reviewer 3 Report

The authors present a new deep learning algorithm to automatically extract and classify EEG features for the diagnosis of alcoholism. They first used discrete wavelet transform to process the standardized data and to improve the signal-to-noise ratio of brain signal. Then, by means of convolutional neural network and bidirectional long-short-term memory network they were able to extract the relevant features from EEG allowing a diagnosis accuracy of alcoholism of more than 99%.

The work is interesting because goes beyond the already used machine learning approaches by employing deep learning algorithms that do not need human intervention.

The results are sound and the conclusions seem supported by the achieved information. The paper is well written and all the sections are well organized.

The only two points that should be considered are:

1)    Missing information about the software/platform that the authors used to design and apply the algorithm. It would be very nice and useful to put some scripts and a block diagram as supplementary materials.

2)    More recent and relevant references should be cited and discussed in the manuscript. There are already similar studies using deep learning algorithms to detect alcoholism like the ref. 29 cited in their paper and letter e in the following list.

a.    10.1007/s11571-016-9416-y

b.    https://doi.org/10.1016/j.bbe.2021.12.009

c.    https://doi.org/10.1016/j.patrec.2019.04.019

d.    https://doi.org/10.1101/2021.06.02.21258251

e.    https://doi.org/10.1080/03772063.2022.2038705

f.      10.1109/BigData52589.2021.9671448

g.    https://doi.org/10.3390/s21165456

Indeed, in my opinion the paper needs minor revisions concerning more details about the used software (and eventually the used scripts) and the addition and discussion of relevant recent references.

Author Response

Thank you for your suggestions, all your suggestions are very important, they have important guiding significance for my thesis writing and scientific research work! we amended the relevant part in manuscript. Some of your questions were answered below:

  • We supplemented our configuration information in the Dataset and configurations section. g., page 3, line 107.
  • We have cited your proposed literature and made minor revisions to the article.

The revised manuscript and responses to each of your suggestions are in the attachments, please check them out, thanks again for your suggestions. Thank you very much for all your help and looking forward to hearing from you soon.

Best regards

Sincerely yours

Mr. Wu
